# Survival Trends in Adults with Out-of-Hospital Cardiac Arrests after Traffic Collisions in Japan: A Population-Based Study

**DOI:** 10.3390/jcm11030745

**Published:** 2022-01-29

**Authors:** Sanae Hosomi, Tetsuhisa Kitamura, Tomotaka Sobue, Ling Zha, Kosuke Kiyohara, Jun Oda

**Affiliations:** 1Department of Traumatology and Acute Critical Medicine, Graduate School of Medicine, Osaka University, 2-15 Yamadaoka, Suita 565-0871, Japan; s-hosomi@hp-emerg.med.osaka-u.ac.jp (S.H.); odajun@gmail.com (J.O.); 2Division of Environmental Medicine and Population Sciences, Department of Social Medicine, Graduate School of Medicine, Osaka University, 2-2 Yamadaoka, Suita 565-0871, Japan; tsobue@envi.med.osaka-u.ac.jp (T.S.); ivy_mist@outlook.com (L.Z.); 3Department of Food Science, Faculty of Home Economics, Otsuma Women’s University, 12 Sanban-cho, Chiyoda-ku, Tokyo 102-8357, Japan; kiyosuke0817@hotmail.com

**Keywords:** traffic collision, mortality, out-of-hospital cardiac arrest, trauma, Japan

## Abstract

The 1-month survival rate from out-of-hospital cardiac arrest (OHCA) of cardiac origin has reportedly improved recently, at ≥5%. However, the characteristics of patients with OHCA after a traffic collision have not been adequately evaluated in Japan. We analyzed the All-Japan Utstein Registry data of 12,577 adult patients aged ≥ 20 years with OHCA due to traffic collisions who were resuscitated by emergency medical service personnel or bystanders and were then transported to medical institutions between 2013 and 2019. Multiple logistic regression analysis was used to assess factors potentially associated with the 1-month survival rate after OHCA. The 1-month survival rate was 1.4% (174/12,577). The proportion of 1-month survival of all OHCAs after a traffic collision origin did not increase significantly (from 1.6% [30/1919] in 2013 to 1.8% [25/1702] in 2019), and the adjusted odds ratio for 1-year increments was 1.04 (95% confidence interval, 0.96–1.12). In multivariate analysis, men who received ventricular fibrillation, pulseless electrical activity, intravenous fluid replacement, or early emergency medical service response and had a traffic collision during daytime had significantly favorable 1-month outcomes. In Japan, the 1-month survival after OHCA of a traffic collision origin was lower than that of a cardiac origin and remains stable.

## 1. Introduction

Traffic collision is a major cause of hospitalization and death globally, resulting in major socioeconomic burden [1,2]. The Global Status Report on Road Safety 2018 shows that the number of deaths due to road accidents has reached 1.35 million annually [3]. The burden is disproportionately borne, with higher trends particularly observed in developing countries. However, even in developed countries, the estimated mortality rates in Europe and the United States are 11.7 and 17 per 100,000 population, respectively [3]. Despite recent advances in driver monitoring and safety assistance control systems, controversy remains as to whether patients’ outcomes with traumatic out-of-hospital cardiac arrests (OHCA) following a traffic collision have improved.

Patients who die after being severely injured due to traffic collisions are usually those who undergo traumatic OHCA at the scene. Patients with traumatic cardiac arrest generally have poorer outcomes than those with non-traumatic cardiac arrest. Even with medical advancements, the ratio of survival to discharge of patients with traumatic OHCA following traffic collision tends to be lower [4,5,6,7]. A traumatic OHCA has a fundamentally different disease process, compared to that of a non-traumatic cardiac arrest, which is predominantly of primary cardiac etiology [6,7]. Thus, the prognostic factors for non-traumatic and traumatic cardiac arrest outcomes tend to be different, and the pathophysiology of traumatic OHCA is different due to the etiology of cardiac arrest. Therefore, other guidelines for termination of resuscitation for OHCA are applicable [8]. Improvements in the chain of survival, including developing public-access defibrillation systems and revisions to cardiopulmonary resuscitation (CPR) guidelines, have led to increased survival after OHCA of cardiac origin in some communities [4,5]; 1-month survival was ≥5% [5,9]. However, the epidemiological characteristics of traumatic OHCAs following traffic collisions has not been sufficiently investigated, as compared to the OHCAs of cardiac origin [10,11,12,13,14]. Therefore, evaluation of the characteristics, trends, and outcomes using detailed traumatic cases and understanding the factors associated with the outcomes are needed to improve survival after traumatic OHCAs following traffic collisions.

In fact, in Japan, the mortality rate associated with traffic collisions has been decreasing [15]. However, the incidence rates of traumatic OHCAs following traffic collisions in Japan, and the outcomes of traumatic OHCAs following traffic collisions are unknown. Herein, we aimed to assess nationwide trends in the incidence and outcomes of traumatic OHCA in adults following traffic collisions in Japan using data from the national registry.

## 2. Materials and Methods

### 2.1. Study Design and Setting

The All-Japan Utstein Registry is a prospective, population-based registry of OHCA that is based on the standardized Utstein style [16,17]. This study enrolled adult patients aged ≥ 20 years who underwent OHCA after a traffic collision before the arrival of emergency medical services (EMS), who were resuscitated by EMS personnel or bystanders and were transported to medical institutions in Japan from 1 January 2013 to 31 December 2019. Evaluation of the detailed causes of non-cardiac OHCA did not begin until 2013. Pediatric patients with OHCA were excluded because the characteristics and outcomes of OHCA differs between children and adults [18,19].

Cardiac arrest is defined as the cessation of cardiac mechanical activity, as confirmed by the absence of signs of circulation [10]. In this registry, cardiac arrests were classified into those of presumed cardiac origin and non-cardiac origin, the latter resulting from cerebrovascular disease, asphyxia, malignant tumors, external causes, drug overuse, anaphylaxis, accidental hypothermia, traffic collision, and other causes. These diagnoses were made clinically by the physician in charge, working in collaboration with the EMS personnel. In this study, patients with traumatic OHCA following a traffic collision were included.

### 2.2. EMS Organization in Japan

Details of the EMS system in Japan have been described previously [4]. In brief, the EMS system is operated by local fire stations. When called, an ambulance is dispatched from the nearest fire station. Emergency services are provided 24 h every day. Most highly trained prehospital emergency care providers are called emergency life-saving technicians (ELSTs). Usually, each ambulance has a crew of three emergency providers, including at least one ELST. They are allowed to insert an intravenous line and an adjunct airway, and to use a semi-automated external defibrillator for patients with OHCA. Since July 2004, specially trained ELSTs have been permitted to perform tracheal intubation, and since April 2006, they have been permitted to administer intravenous epinephrine. Do-not-resuscitate orders or living wills are not generally accepted in Japan. EMS providers are not permitted to terminate resuscitation in the field. Therefore, almost all patients with OHCA who are treated by EMS personnel are transported to a hospital and enrolled in the All-Japan Utstein Project, excluding those with decapitation, incineration, decomposition, rigor mortis, or dependent cyanosis.

The use of an automated external defibrillator (AED) by citizens has been legally permitted since July 2004. All EMS providers perform cardiopulmonary resuscitation (CPR) according to the Japanese CPR guidelines [20]. In Japan, approximately 2 million citizens per year participate in community CPR programs, which include training in chest compression, mouth-to-mouth ventilation, and AED use [4,5].

### 2.3. Data Collection and Quality Control

Data were prospectively collected using a form that included data recommended in the Utstein-style reporting guidelines for cardiac arrests [16,17]. Data on patient age, sex, type of bystander witness status, first recorded cardiac rhythm, life support by EMS personnel (i.e., use of advanced life support (ALS) devices and insertion of an intravenous line), time course of resuscitation, epinephrine administration, prehospital return of spontaneous circulation (ROSC), and 1-month survival rates were obtained. Data on EMS call receipt times, arrival of the ambulance at the scene of the accident, contact with patients, initiation of CPR, defibrillation performed by EMS personnel, and arrival at the hospital were recorded using the clock of each EMS system. In cases of shock delivery by bystanders using a public-access AED, the patient’s first recorded rhythm was regarded as ventricular fibrillation (VF) or pulseless ventricular tachycardia (VT). Information on the type of bystander CPR was obtained through observation and interviews with the bystander, which was performed by EMS personnel before leaving the scene of the accident. The data forms were completed by EMS personnel in cooperation with treating physicians. The data were integrated into the All-Japan Utstein Registry database server and were logically checked by the computer system. In cases of incomplete data forms, the Fire and Disaster Management Agency requested the provision of missing data from the respective fire stations.

All survivors who experienced OHCA were followed-up for up to 1 month after the event by the EMS personnel in charge. One-month neurological outcomes were determined by the physician responsible for treating the patient, using the cerebral performance category (CPC) scale that is measured as follows: category 1, good cerebral performance; category 2, moderate cerebral disability; category 3, severe cerebral disability; category 4, coma or vegetative state; and category 5, death [16,17].

### 2.4. Outcome Measures

The main outcome measure was the 1-month survival rate. The secondary outcome measures included prehospital ROSC and a 1-month survival rate with neurologically favorable outcomes, defined as CPC categories 1 or 2 [16,17].

### 2.5. Statistical Analysis

Categorical variables are presented as counts with proportions, and the χ^2^ test was used to evaluate the differences between the two groups. Continuous variables are presented as medians with interquartile ranges (IQRs), and the Wilcoxon Mann–Whitney U test was used to evaluate differences between the two groups.

The age-standardized annual incidence of OHCAs after traffic collisions was calculated by the direct method using the 2013–2019 population data from the Statistics Bureau of Japan and the 1985 Japanese model population [21,22]. Annual trends were assessed using linear trend tests. Multiple logistic regression analysis was used to assess factors associated with 1-month survival, prehospital ROSC, neurologically favorable outcomes, and adjusted odds ratios (ORs) and their 95% confidence intervals (CIs) were calculated. As potential confounders, factors that were biologically essential and considered to be associated with clinical outcomes were included in the multivariable analyses [23]. These variables included age (20–64/65–74/≥75 years old), sex, witness status (none/witnessed by bystanders), first documented rhythm (VF/pulseless VT/pulseless electrical activity (PEA)/asystole), bystander CPR status (any CPR/no CPR), advanced airway management (AAM) (endotracheal intubation /supraglottic airway (SGA)/none), intravascular fluid (yes/no), epinephrine (yes/no), EMS response time (call to contact with patients), contact with patients to hospital arrival, daytime (9:00 a.m.–4:59 p.m.) (yes/no), weekend/holiday (yes/no), and year of cardiac arrest. In the subgroup analysis, we conducted a multivariate analysis of 1-month survival from OHCAs after dividing the patients into three groups based on their age: 20–64, 65–74, and ≥75 years.

All statistical analyses were conducted using STATA (version 16; StataCorp LP, College Station, TX, USA). All tests were two-tailed, and *p*-values of <0.05 were considered statistically significant.

### 2.6. Ethics Approval

This manuscript complies with the STROBE statement for the reporting of cohort and cross-sectional studies [24]. The study design was approved by the Ethics Committee of the Osaka University Graduate School of Medicine (approval number: 14147). The requirement for written informed consent was waived, owing to the retrospective nature of the study. Personal identifiers were not included in the Utstein records.

## 3. Results

### 3.1. Eligible Patients

Figure 1 shows an overview of the study patients based on the Utstein template. Data of a total of 866,214 adult patients with cardiac arrest were documented during these seven years. Resuscitation was attempted in 845,632 patients. After excluding 67,590 victims who were witnessed by EMS (experienced cardiac arrest after EMS arrival) and 3772 in unknown witnessed cases, 774,270 patients (280,989 bystander-witnessed arrests and 493,281 non-witnessed arrests) were included in the analysis. Among these cases of cardiac arrest, 12,843 were due to traffic collisions. We could not obtain information on the first cardiac rhythm and bystander CPR for 266 (2.07%) patients. Finally, the remaining 12,577 patients were eligible for our study.

### 3.2. Annual Incidence of Traumatic OHCA Following Traffic Collision

The age-standardized annual incidence rates of traumatic OHCA following traffic collision per 100,000 persons were calculated over time (Figure 2). The incidence rate of traumatic OHCA following traffic collision did not decrease from 2013 to 2019 in all age groups except among the elderly, aged ≥ 75 years (*p* = 0.148 in 20–64 years, *p* = 0.052 in 65–74 years, and *p* < 0.001 in ≥75 years).

### 3.3. Baseline Characteristics

The characteristics of patients and EMS personnel involved in traumatic OHCA following traffic collision are shown in Table 1, and their outcomes are shown in Table 2. The median age of all patients with traumatic OHCA following traffic collision was 65 years (range, 45–76) in 2013 and 66 years (range, 46–78) in 2019 (*p* = 0.149) and the proportion of men was 68.2% in 2013 and 69.5% in 2019 (*p* = 0.228), which were stable over time.

The number of cases with a witness (61.9% in 2013 and 64.6% in 2019; *p* < 0.001) significantly increased, along with the frequency of bystander CPR (20.5% in 2013 and 27.0% in 2019; *p* < 0.001). The first rhythm was commonly a non-shockable rhythm (pulseless electrical activity and asystole), and VF rhythm decreased from 3.3% in 2013 to 1.0% in 2019 (*p* < 0.001). While the proportion of AAM was stable, the rate of intravascular fluid and epinephrine administration increased (*p* < 0.001). Most cases of traumatic OHCA following traffic collision occurred during nighttime and on weekends/holidays.

### 3.4. Outcomes

In all patients (N = 12,577), the rates of 1-month survival, prehospital ROSC, and neurologically favorable outcomes with CPC categories 1 or 2 were 1.4% (n = 174), 4.4% (n = 554), and 0.3% (n = 44), respectively. The proportion of 1-month survival rates of traumatic OHCA following traffic collision did not significantly increase (from 1.6% (30/1919) in 2013 to 1.5% (25/1647) in 2019) and the adjusted OR for annual increment was 1.04 (95% CI, 0.96–1.12). The adjusted OR for increases in the 1-month survival rates, prehospital ROSC rates, and neurologically favorable outcomes after traumatic OHCA following traffic collision were almost similar to those of the 1-month survival rate (adjusted OR, 1.01; 95% CI, 0.97–1.06; adjusted OR, 1.12; 95% CI, 0.96–1.30; respectively).

Age-related subgroup analyses after dividing patients into three groups (20–64 years, 65–74 years, and ≥75 years) are shown in Table 3. The 1-month survival trend of patients with traumatic OHCA following traffic collision was almost stable during the study period (20–64 years: adjusted OR, 1.11; 95% CI, 0.99–1.24; 65–74 years: adjusted OR, 1.05; 95% CI, 0.91–1.22; and ≥75 years: adjusted OR, 0.94; 95% CI, 0.80–1.09).

### 3.5. Factors Related to Mortality

Table 4 shows the factors contributing to 1-month survival after traumatic OHCA following a traffic collision. With regard to 1-month survival, male sex (adjusted OR, 1.56; 95% CI, 1.08–2.25), VF as the first documented rhythm (adjusted OR, 15.32; 95% CI, 8.08–29.07), PEA as the first documented rhythm (adjusted OR, 5.77; 95% CI, 3.92–8.49), intravenous fluid levels (adjusted OR, 3.00; 95% CI, 1.99–4.50), early EMS response time (adjusted OR for 1-min-increment 0.96; 95% CI, 0.93 to 1.00), and daytime (adjusted OR, 1.89; 95% CI, 1.39–2.58) were associated with favorable outcomes. However, AAM by SGA (adjusted OR, 0.50; 95% CI, 0.33–0.76) and epinephrine (adjusted OR, 0.42; 95% CI, 0.25–0.69) were not associated with favorable outcomes.

## 4. Discussion

This study described the actual situation of the incidence and outcomes of traumatic OHCAs in adults following traffic collisions, thereby providing valuable information to appropriately manage patients and improve survival. This study, based on the extensive OHCA registry of Japan, showed that the 1-month survival after a traumatic OHCA following traffic collision was 1.4%, and that the survival trends did not improve year-by-year. To further improve survival after OHCA, attention should be paid to the epidemiological characteristics of traumatic OHCA following a traffic collision, as is paid to OHCAs of cardiac origin.

More than 120,000 Japanese individuals experience OHCA annually [4,5]. Although the survival rate of OHCA is improving, ≥5% of individuals survive 1 month after experiencing OHCA [4,5,9,25]. We found that patients with traumatic OHCA have an extremely low survival rate, possibly due to the changing epidemiology of patients with traumatic OHCA following traffic collisions in older patients with a higher number of comorbidities. Globally, Japan is now one of the few countries with a super-aged population. However, the annual incidence rates of the elderly, aged ≥ 75 years, with traumatic OHCA following traffic collisions decreased during the study period.

Survival after traumatic OHCA is rare, even with maximal resuscitative efforts. Most importantly, efforts should focus on the prevention of traumatic OHCA following traffic collisions because most cases are preventable [1,2,3,4]. To reduce trauma-related deaths caused by motor vehicle collisions (MVCs), the Japanese Road Traffic Act was revised in June 2002, and it imposes severe fines for traffic offenses. For example, the incidence of fatal collisions caused by drunk drivers has decreased since then. It was reported that the age-standardized incidences per 100,000 persons ranged from 1.6 in 2005 to 1.4 in 2011 (*p* = 0.229) and the unadjusted 1-month survival rate ranged from 0.9% (1/116) in 2005 to 2.6% (3/115) in 2011 (*p* = 0.027) in Osaka, Japan [21]. However, in our study, this rate seems to be plateauing in our study, thereby warranting further improvement. There are three approaches to road safety: the traditional approach, which focuses on human errors and road users; the systemic approach, which includes sustainable safety and safe systems; and the vision zero, which is a multi-national road traffic safety project that aims to achieve a highway system with no fatalities or serious injuries following traffic collision. More recently, traffic collisions caused by older individuals have become a serious social problem, and a cognitive test for the license renewal procedure is required for drivers aged ≥ 75 years to reduce MVCs in Japan. The test has been obligatory since June 2009 [26], helping more elderly people to voluntarily return their license. Furthermore, many automobile manufacturers strive to improve safe driving support systems and devices, such as improved occupant recognition performance of the driver monitoring system, vehicle motion control technology, and the introduction of “connected safety” technologies, such as the Advanced Automatic Collision Notification and infrastructure coordination.

Our study showed the proportion of bystander CPR for traumatic OHCA cases following a traffic collision in Japan might be lower than that of OHCAs of cardiac origin [4]. There might be differences in the proficiency of bystander basic life support (BLS) procedures between cardiac and traumatic OHCA cases. Some traumatic deaths following MVCs occur due to severe brain injury or hemorrhage shock [27]. In serious trauma cases, wherein the patients experience a cardiac arrest at the scene, the effectiveness of BLS might be clinically limited. Furthermore, EMS response time was an independent predictor of favorable outcomes in the multivariate analysis. As recommended in the CPR guidelines [19,28], activating EMS plays a key role in the “chain of survival”.

In Japan, EMS personnel are not allowed to perform certain advanced interventions (for example, surgical airway, chest drain, or intraosseous access). They are not allowed to insert chest tubes, even when pneumothorax is suspected. This means that EMS personnel in Japan are permitted to perform only intravenous catheter insertion and fluid infusion, intravenous epinephrine administration, and endotracheal intubation. EMS personnel need permission or instructions from a medical director in each municipality to perform ALS on a case-to-case basis, which may prevent EMS personnel from providing timely treatment. In this study, very few patients received prehospital ALS procedures (i.e., intravenous catheter insertion, epinephrine administration, or AAM by SGA). However, in our study, intravenous access was associated with a reduction in hospital mortality among patients. This may be because traumatic OHCA is mostly due to blood loss, pneumothorax, or pericardial tamponade [27]. The etiology of cardiac arrest among the patients included in this study was blunt trauma [29]. Conversely, the multivariate analysis also showed that epinephrine and AAM by SGA were not effective. Indeed, they seemed to be associated with worse outcome, as suggested by previous observational studies [30,31]. However, we considered that there could be an inversion phenomenon of cause and effect; it would be difficult to assess the effect of ALS measures in this observational study because the EMS personnel in Japan could provide ALS measures only for OHCA patients who did not respond to BLS. Recently, some studies suggested that earlier epinephrine administration and endotracheal intubation contributed to improving outcomes after OHCA [25]. These earlier interventions would be surrogate parameters for a team that works faster and/or is more experienced overall. Further investigations by other cohorts and randomized controlled trials are needed to confirm these associations. Another effective option may be to permit EMS personnel to perform ALS based on their own judgment and/or expand the range of available procedures that EMS personnel can learn, practice, and perform. In addition, verifying these effects on prehospital emergency care and in-hospital treatment is essential [32].

Recent reports investigated the association between the time and day of the week with outcomes from adult OHCAs [33,34] and in-hospital cardiac arrests [35]. The survival rates of those who experienced cardiac arrests during nights and on weekends was found to be lower than those who experienced them during the day and on weekdays. This study demonstrated that traumatic OHCA following traffic collisions occurs more often at nighttime than during daytime and more often on weekends/holidays than on weekdays. Furthermore, the 1-month survival rate of traumatic OHCA cases following traffic collisions was significantly lower during nights than during days. This poorer outcome might be due to a longer reporting time of the accident or due to understaffed hospitals during the night. In addition, factors such as male sex were also independent predictors of better outcomes after traumatic OHCAs following traffic collision in a multivariate analysis. Our study is consistent with the findings of previous studies, although the reason is unknown [36].

This study has some inherent limitations. Our Utstein-based data did not include data on in-hospital treatments, such as trauma care (fluid resuscitation, emergent thoracotomy, and aortic cross-clamping) [27], post-cardiac arrest care [37], and hospital staffing. Furthermore, the autopsy data were not obtained because our data were collected in accordance with the Utstein-style guidelines by EMS personnel. Therefore, our study could not identify the detailed origin of the presumed OHCAs nor estimate OHCAs due to a medical origin (e.g., pulmonary embolism or cardiac arrhythmias, such as VF/pVT) in those caused by traffic injuries. Second, our results might not be fully applicable to other countries, including the United States and Europe, which have different EMS and medical systems. Currently, procedures by paramedics on the scene are limited in Japan; there is a need for them to be allowed to perform other procedures (e.g., insertion of chest tubes) to reduce the mortality rates of OHCA due to traffic collision. Therefore, further investigations of other cohorts are needed to confirm these associations and to address their generalizability. Third, there might be unmeasured confounding factors that might have influenced the association between traumatic OHCA following traffic collisions and outcomes. In addition, we could not identify the most severely injured organ from the registry. We did not know whether the victims were pedestrians, cyclists, motorcyclists, or drivers/passengers. This study does not include the COVID-19 pandemic period, and the results may change if a longer observation period is considered [38]. Finally, as with all epidemiological studies, data integrity, validity, and ascertainment bias are potential limitations. The use of uniform data collection based on Utstein-style guidelines for reporting cardiac arrest, a large sample size, and a population-based design to cover all known OHCAs in Japan was intended to minimize these potential sources of bias.

## 5. Conclusions

The large OHCA registry of Japan indicated that the 1-month survival rate after traumatic OHCAs following traffic collisions is lower than those of cardiac origin. Survival trends did not improve with year, and survival did not differ by age. The prevention of traumatic OHCAs and a greater range of prehospital treatment methods for traumatic OHCAs following traffic collision are warranted to improve survival.

## Figures and Tables

**Figure 1 jcm-11-00745-f001:**
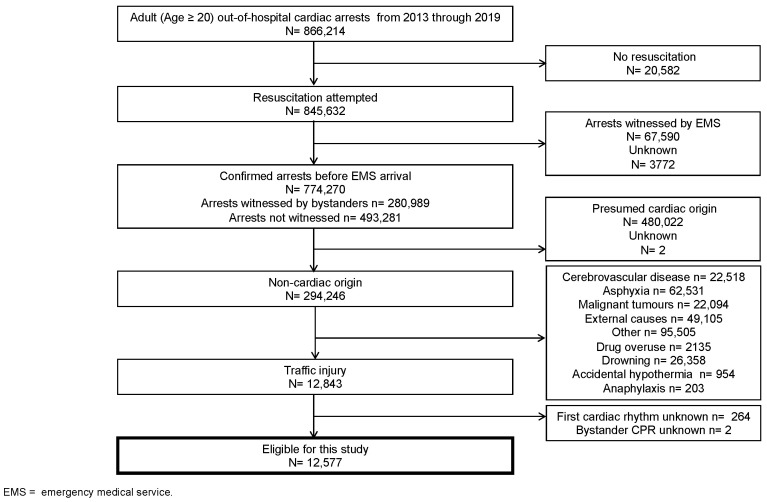
Flow chart of the study population.

**Figure 2 jcm-11-00745-f002:**
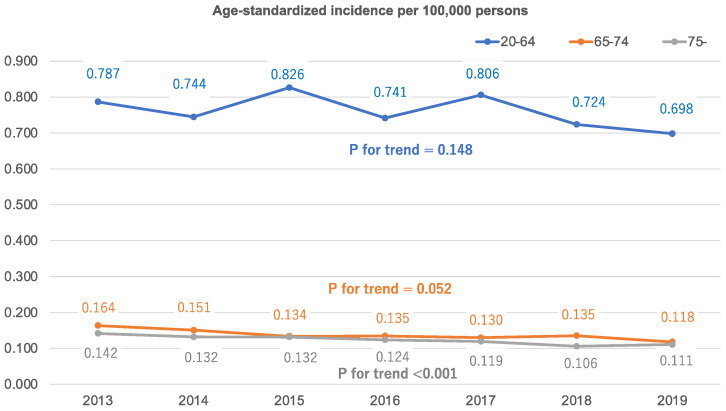
Age-standardized incidences of traumatic out-of-hospital cardiac arrests following traffic collisions.

**Table 1 jcm-11-00745-t001:** Characteristics of patients and EMS involved in traumatic out-of-hospital cardiac arrests after traffic collisions.

		2013	2014	2015	2016	2017	2018	2019	*p* for Trend
		N = 1919	N = 1824	N = 1871	N = 1775	N = 1839	N = 1702	N = 1647	
Age, year, median (IQR)		65 (45–76)	65 (47–77)	63 (44–76)	66 (45–78)	64 (45–77)	65 (46–77)	66 (46–78)	0.149
Age group, n (%)	Adults aged 20–64 years	956 (49.8%)	891 (48.8%)	968 (51.7%)	857 (48.3%)	938 (51.0%)	839 (49.3%)	795 (48.3%)	0.503
	Elderly aged 65–74 years	405 (21.1%)	392 (21.5%)	355 (19.0%)	356 (20.1%)	346 (18.8%)	364 (21.4%)	315 (19.1%)	0.190
	Elderly aged ≥ 75 years	558 (29.1%)	541 (29.7%)	548 (29.3%)	562 (31.7%)	555 (30.2%)	499 (29.3%)	537 (32.6%)	0.059
Sex, n (%)	Male	1309 (68.2%)	1224 (67.1%)	1298 (69.4%)	1239 (69.8%)	1264 (68.7%)	1173 (68.9%)	1145 (69.5%)	0.228
Witness, n (%)	Arrests witnessed by bystanders	1187 (61.9%)	1150 (63.0%)	1159 (61.9%)	1173 (66.1%)	1233 (67.0%)	1135 (66.7%)	1064 (64.6%)	<0.001
First documented rhythm, n (%)	VF/pVT	63 (3.3%)	30 (1.6%)	23 (1.2%)	23 (1.3%)	31 (1.7%)	26 (1.5%)	17 (1.0%)	<0.001
	PEA	665 (34.7%)	613 (33.6%)	619 (33.1%)	631 (35.5%)	631 (34.3%)	631 (37.1%)	601 (36.5%)	0.023
	Asystole	1191 (62.1%)	1181 (64.7%)	1229 (65.7%)	1121 (63.2%)	1177 (64.0%)	1045 (61.4%)	1029 (62.5%)	0.264
Bystander CPR, n (%)		393 (20.5%)	418 (22.9%)	442 (23.6%)	437 (24.6%)	450 (24.5%)	451 (26.5%)	444 (27.0%)	<0.001
Advanced airway management, n (%)	ETI	57 (3.0%)	54 (3.0%)	74 (4.0%)	66 (3.7%)	74 (4.0%)	59 (3.5%)	60 (3.6%)	0.158
	SGA	514 (26.8%)	492 (27.0%)	477 (25.5%)	451 (25.4%)	443 (24.1%)	418 (24.6%)	428 (26.0%)	0.095
	Non	1348 (70.2%)	1278 (70.1%)	1320 (70.6%)	1258 (70.9%)	1322 (71.9%)	1225 (72.0%)	1159 (70.4%)	0.308
Intravascular fluid, n (%)		440 (22.9%)	438 (24.0%)	411 (22.0%)	450 (25.4%)	505 (27.5%)	468 (27.5%)	482 (29.3%)	<0.001
Epinephrine, n (%)		245 (12.8%)	263 (14.4%)	236 (12.6%)	289 (16.3%)	325 (17.7%)	321 (18.9%)	356 (21.6%)	<0.001
Call to contact with a patient by EMS, min, median (IQR)		9 (7–12)	9 (7–13)	9 (7–13)	9 (7–13)	9 (7–13)	10 (7–13)	9 (7–13)	<0.001
Contact to hospital arrival, min, median (IQR)		24 (17–34)	24 (17–34)	24 (17–34)	24 (18–34)	25 (18–35)	25 (18–35)	24 (18–35)	0.006
Daytime, n (%)		652 (34.0%)	629 (34.5%)	634 (33.9%)	645 (36.3%)	651 (35.4%)	594 (34.9%)	569 (34.5%)	0.455
Weekend/Holiday, n (%)		1409 (73.4%)	1281 (70.2%)	1345 (71.9%)	1243 (70.0%)	1347 (73.2%)	1243 (73.0%)	1165 (70.7%)	0.830

EMS, emergency medical services; ETI, endotracheal intubation; IQR, interquartile range; PEA, pulseless electrical activity; SGA, supraglottic airway; VF, ventricular fibrillation; pVT, pulseless ventricular tachycardia.

**Table 2 jcm-11-00745-t002:** Annual trends in the primary and secondary outcomes of traumatic out-of-hospital cardiac arrests after traffic collisions.

	Total	2013	2014	2015	2016	2017	2018	2019	
	N = 12,577	N = 1919	N = 1824	N = 1871	N = 1775	N = 1839	N = 1702	N = 1647	
**One-month survival, n (%)**	174 (1.4%)	30 (1.6%)	16 (0.9%)	26 (1.4%)	25 (1.4%)	22 (1.2%)	30 (1.8%)	25 (1.5%)	OR for 1-year increment
Crude OR		reference	0.56	0.89	0.90	0.76	1.13	0.97	1.04
95%CI			(0.30–1.03)	(0.52–1.51)	(0.53–1.54)	(0.44–1.33)	(0.68–1.88)	(0.57–1.66)	(0.96–1.12)
Adjusted OR		reference	0.59	1.01	0.98	0.77	1.18	1.06	1.04
95%CI			(0.32–1.10)	(0.59–1.73)	(0.57–1.69)	(0.43–1.36)	(0.70–1.99)	(0.61–1.83)	(0.96–1.12)
**Prehospital ROSC, n (%)**	554 (4.4%)	92 (4.8%)	61 (3.3%)	75 (4.0%)	85 (4.8%)	76 (4.1%)	76 (4.5%)	89 (5.4%)	OR for 1-year increment
Crude OR		reference	0.69	0.83	1.00	0.86	0.93	1.13	1.03
95%CI			(0.49–0.96)	(0.61–1.13)	(0.74–1.35)	(0.63–1.17)	(0.68–1.27)	(0.84–1.53)	(0.99–1.08)
Adjusted OR		reference	0.66	0.88	0.91	0.76	0.83	1.01	1.01
95%CI			(0.47–0.93)	(0.64–1.21)	(0.66–1.25)	(0.55–1.06)	(0.60–1.14)	(0.74–1.38)	(0.97–1.06)
**Neurological favorable outcome with CPC categories 1 or 2, n (%)**	44 (0.3%)	7 (0.4%)	3 (0.2%)	9 (0.5%)	5 (0.3%)	2 (0.1%)	11 (0.6%)	7 (0.4%)	OR for 1-year increment
Crude OR		reference	0.45	1.32	0.77	0.30	1.78	1.17	1.08
95%CI			(0.12–1.74)	(0.49–3.55)	(0.24–2.44)	(0.06–1.43)	(0.69–4.59)	(0.41–3.33)	(0.93–1.25)
Adjusted OR		reference	0.52	1.65	0.99	0.37	2.11	1.62	1.12
95%CI			(0.13–2.05)	(0.60–4.56)	(0.31–3.22)	(0.07–1.80)	(0.79–5.62)	(0.56–4.81)	(0.96–1.30)

OR, odds ratio; CI, confidence Interval; ROSC, return of spontaneous circulation; CPC, cerebral performance category.

**Table 3 jcm-11-00745-t003:** One-month survival of patients with traumatic out-of-hospital cardiac arrests after traffic accidents over time by age group.

	Total	2013	2014	2015	2016	2017	2018	2019	
**20–64, n/N**	86/6244	13/956	7/891	15/968	10/857	10/938	18/839	13/795	
	1.38%	1.36%	0.79%	1.55%	1.17%	1.07%	2.15%	1.64%	OR for 1-year increment
Crude OR		reference	0.57	1.14	0.86	0.78	1.59	1.21	1.08
95%CI			(0.23–1.45)	(0.54–2.41)	(0.37–1.96)	(0.34–1.79)	(0.77–3.27)	(0.56–2.62)	(0.97–1.20)
Adjusted OR		reference	0.75	1.50	1.10	0.96	1.87	1.67	1.11
95%CI			(0.29–1.94)	(0.69–3.26)	(0.47–2.60)	(0.41–2.27)	(0.88–3.97)	(0.74–3.75)	(0.99–1.24)
**65–74, n/N**	47/2533	6/405	7/392	7/355	5/356	8/346	7/364	7/315	
	1.86%	1.48%	1.79%	1.97%	1.40%	2.31%	1.92%	2.22%	OR for 1-year increment
Crude OR		reference	1.21	1.34	0.95	1.57	1.30	1.51	1.06
95%CI			(0.40–3.63)	(0.45–4.02)	(0.29–3.13)	(0.54–4.58)	(0.43–3.92)	(0.50–4.54)	(0.91–1.22)
Adjusted OR		reference	1.08	1.53	1.00	1.30	1.37	1.47	1.05
95%CI			(0.35–3.31)	(0.50–4.69)	(0.30–3.40)	(0.42–4.00)	(0.44–4.24)	(0.48–4.53)	(0.91–1.22)
**≥75, n/N**	41/3800	11/558	2/541	4/548	10/562	4/555	5/499	5/537	
	1.08%	1.97%	0.37%	0.73%	1.78%	0.72%	1.00%	0.93%	OR for 1-year increment
Crude OR		reference	0.18	0.37	0.90	0.36	0.50	0.47	0.94
95%CI			(0.04–0.84)	(0.12–1.16)	(0.38–2.14)	(0.11–1.14)	(0.17–1.46)	(0.16–1.35)	(0.80–1.09)
Adjusted OR		reference	0.18	0.37	0.81	0.38	0.51	0.44	0.94
95%CI			(0.04–0.81)	(0.12–1.19)	(0.33–1.95)	(0.12–1.23)	(0.17–1.51)	(0.15–1.28)	(0.80–1.09)

OR, odds ratio; CI, confidence interval.

**Table 4 jcm-11-00745-t004:** Factors associated with primary outcomes.

		All (N)	1 Month Survival (N)	(%)	Crude OR	95% CI	Adjusted OR	95% CI
Age group, n (%)	Adults aged 18–64 years	6244	86	1.38%	(reference)		(reference)	
	Elderly aged 65–74 years	2533	47	1.86%	1.35	(0.95–1.94)	1.26	(0.87–1.83)
	Elderly aged ≥ 75 years	3800	41	1.08%	0.78	(0.54–1.14)	0.70	(0.47–1.03)
Sex, n (%)	Female	3925	42	1.07%	(reference)		(reference)	
	Male	8652	132	1.53%	1.43	(1.01–2.03)	1.56	(1.08–2.25)
Witness, n (%)	Arrests witnessed by bystanders	8101	117	1.44%	1.14	(0.83–1.56)	1.01	(0.73–1.42)
	Arrests not witnessed	4476	57	1.27%	(reference)		(reference)	
First documented rhythm, n (%)	VF/pVT	213	15	7.04%	16.25	(8.77–30.09)	15.32	(8.08–29.07)
	PEA	4391	122	2.78%	6.13	(4.24–8.87)	5.77	(3.92–8.49)
	Asystole	7973	37	0.46%	(reference)		(reference)	
Bystander CPR, n (%)	No	9542	138	1.45%	(reference)		(reference)	
	Yes	3035	36	1.19%	0.82	(0.57–1.18)	0.70	(0.47–1.02)
Advanced airway management, n (%)	ETI	444	9	2.03%	1.34	(0.68–2.66)	1.06	(0.50–2.23)
	SGA	3223	30	0.93%	0.61	(0.41–0.91)	0.50	(0.33–0.76)
	None	8910	135	1.52%	(reference)		(reference)	
Intravascular fluid, n (%)	No	9383	111	1.18%	(reference)		(reference)	
	Yes	3194	63	1.97%	1.68	(1.23–2.30)	3.00	(1.99–4.50)
Epinephrine, n (%)	No	10,542	144	1.37%	(reference)		(reference)	
	Yes	2035	30	1.47%	1.08	(0.73–1.61)	0.42	(0.25–0.69)
Call to contact with a patient by EMS, min, median (IQR)					0.93	(0.89–0.96)	0.96	(0.93–1.00)
Contact to hospital arrival, min, median (IQR)					0.99	(0.98–1.00)	1.00	(0.99–1.01)
Daytime, n (%)	No	8203	91	1.11%	(reference)		(reference)	
	Yes	4374	83	1.90%	1.72	(1.28–2.33)	1.89	(1.39–2.58)
Weekend/Holiday, n (%)	No	11,992	168	1.40%	(reference)		(reference)	
	Yes	585	6	1.03%	0.73	(0.32–1.65)	0.60	(0.24–1.47)

PEA, pulseless electrical activity; SGA, supraglottic airway; ETI, endotracheal intubation; IQR, interquartile range; VF, ventricular fibrillation; OR, odds ratio; CI, confidence interval; CPR, cardiopulmonary resuscitation; EMS, emergency medical services.

## Data Availability

The data that support the findings of this study are available from the All-Japan Utstein Registry; restrictions apply to the availability of these data, which were used under license for the current study, and are therefore, not publicly available. Data are, however, available from the authors upon reasonable request and with permission from the All-Japan Utstein Registry.

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
