# Peer review of "Survival Trends in Adults with Out-of-Hospital Cardiac Arrests after Traffic Collisions in Japan: A Population-Based Study"

_jcm, 2022, doi:10.3390/jcm11030745_

Round 1

Reviewer 1 Report

Dear ladies and gentlemen,

thank you very much for the possibility to review this very good scientific paper.

I have some questions and suggestions which are stated here:

Lines 68/69: Were patients excluded if arrest was post arrival?

Lines 90/91: What are ELSTs not allowed to do? You could mention that they are not allowed to insert chest tubes in cases of suspicion of pneumothorax as you state in -> ll. 281/282. From my point of view this is important for this topic.

Lines 265/266: required for drivers aged > 75 years to reducing MVCs in Japan?. (is this a requirement in Japan?)

Lines 272-276: how do you differentiate between cardiac arrest which leads to MVC vs. trauma leads to cardiac arrest. This seems to be an important question for this paper since patients who develop acute cardiologic illnesses, e.g. PE or cardiac arrhythmias such as V-fib, can cause an MVC. In such a case it can happen that the patient suffers mild traumatic injuries only and therefore should be treated as a cardiac arrest based on the underlying condition (in these examples treatment of PE or V-fib).

Lines 285/286: permission given on a case-to-case basis or generally?

Line 290: “mostly due to blood loss?” Not other causes such as pneumothorax, pericardial tamponade etc.? Do you have a citation for this?

Lines 298/299: could “earlier endotracheal intubation” and “earlier epinephrine” be surrogate parameters for a team which works faster and/or is more experienced overall?

Lines 320/321: Should there be a discussion that paramedics should be allowed to perform other procedures in these cases, e.g. insertion of chest tubes? Or should there be emergency physicians on the scene performing it in these cases? Could this mean an advantage to EMS systems which operate with emergency physicians who work pre-clinically?

I would be willing and delighted to review a revised version.

Sincerely

Reviewer 2 Report

I congratulate the authors to this nice work. They study the important topic of outof hospital reanimation following car accidents and showed an improved survival rate when compared with not-traumatic requriements off caridac arrest. I have some general questions: have you seen patients who had a car accident following a crdiac arrest? is their survival comparable to the trauma or the heart patients? I am missing the definition of "neurological favorable outcome " in Table 2. How was that assessed? I would suggest to discarde the total number to improve readability: Instead 1.4% (174/12,577), write 1.4%(n = 174), or 174 (1.4%).
